# Network Embedding using Hierarchical Feature Aggregation

**Ujjawal Soni, Mohan Bhambhani & Mitesh M. Khapra**
Department of Computer Science & Engineering
Indian Institute of Technology Madras
{ujjawals,mohanpb,miteshk}@cse.iitm.ac.in

## Abstract

Graph convolutional networks and its variants are the state-of-the-art methods for learning node embeddings in a graph-structured data. However, these existing approaches fail to capture the neighborhood information efficiently beyond a certain depth from the node. In this work, we propose a novel hierarchical feature aggregation approach which explictly aggregates the feature information from different depths of a node's neighborhood using an LSTM model. Proposed model gives promising results on four real-world datasets as compared to state-of-the-art methods.

## 1 Introduction and Related Work

Recent years have seen a surge in network embedding approaches, [Perozzi et al. (2014); Grover & Leskovec (2016); Wang et al. (2016); Tang et al. (2015)], wherein we learn to map the nodes in a network to a low-dimensional vector space preserving the network structure as well as the node feature information. The learned dense node embeddings are then used with of-the-shelf machine learning techniques to solve downstream network tasks like node classification, node clustering, link prediction, etc.

Inspired from the success of Convolutional Neural Networks (CNNs) for solving challenging problems like image classification in Computer Vision to Machine Translation in Natural Language Processing, there have been several works in extending CNNs to arbitrarily structured networks [Duvenaud et al. (2015); Defferrard et al. (2016); Niepert et al. (2016); Kipf & Welling (2017); Hamilton et al. (2017); Schlichtkrull et al. (2017); Chen & Zhu (2017)]. These convolution based approaches for network embedding leverage the feature information of a node as well as its (full [Kipf & Welling (2017)] or partial [Hamilton et al. (2017)]) neighborhood, by learning appropriate feature aggregation functions. Graph Convolutional Networks (GCN) [Kipf & Welling (2017)] look at the complete 1-hop neighborhood around the node for aggregation and multiple depths of the model helps capture higher-order information. Contrary to looking at full neighborhood, GraphSAGE [Hamilton et al. (2017)] looks at partially sampled neighborhood around the node.

GCN, GraphSAGE (and their variants) have two key limitations:

- They fail to capture information beyond second-order neighborhood of a node. Since, GCN looks at full neighborhood of a node, going beyond second-order would require exponentially more computations and is practically not feasible on many real-world networks. While GraphSAGE, despite considering sampled neighborhood, does not gain much performance going beyond second-order depth.
- Both GraphSAGE and GCN iteratively propagate neighborhood features to the node, i.e. higher-depth information is propagated via nodes at lower-depth. Thus, the propagation from nodes at higher-depth proximity gets averaged-out multiple times before reaching the center node. This causes information morphing at each step.

In order to tackle these challenge, we propose a hierarchical feature aggregation approach where we explicitly learn neighborhood feature aggregators at different depths from the node. Information gathered at different depths from the node are then aggregated using an LSTM model, ensuring structured information flow from higher-depths towards the nodes.

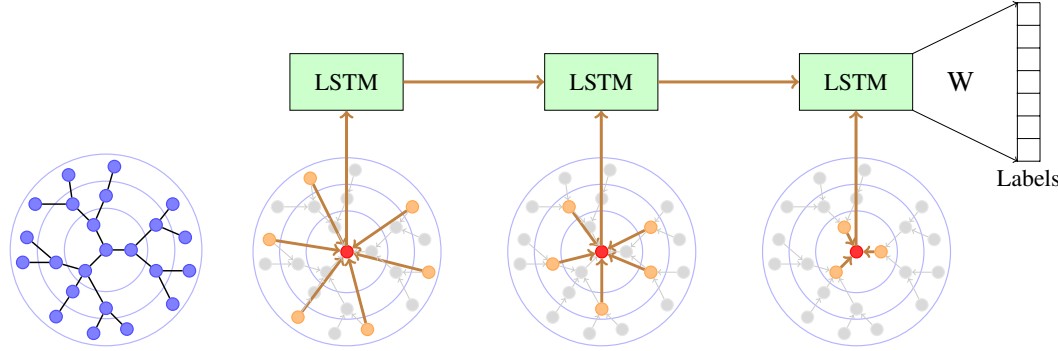

(a) A sample toy network

(b) Visual overview of our approach to encode higher-order feature information for node embedding.

Figure 1: Our hierarchical feature aggregation model where we initially aggregate features at different depth of a node. These aggregated depth embeddings of a node are then fed to an LSTM model followed by a fully-connected layer for the down-stream task. In the example network, we perform feature aggregation till three depths of a node.

## 2 PROPOSED MODEL

### 2.1 NOTATIONS

We denote a network by $\mathcal{G} = (\mathcal{V}, \mathcal{E}, \mathcal{X}, \mathcal{L})$, where $\mathcal{V}$ is the node set, $\mathcal{E}$ is the set of edges between the nodes, $\mathcal{X} = \{x_v \in R^F, \forall v \in \mathcal{V}\}$ denote the node features, and $\mathcal{L}$ denotes the labels for the nodes. The neighborhood of a node $v$ at distance $k$ from $v$ is denoted by $\mathcal{N}_v^{(k)}$. The learned embeddings are denoted by $\{h_v, \forall v \in V\}$ and embeddings w.r.t. depth $k$ for each node are denoted by $\{h_v^{(k)}, \forall v \in V\}$.

### 2.2 HIERARCHICAL FEATURE AGGREGATION MODEL

The proposed model learns the node embeddings via a hierarchical aggregation framework. Firstly, using aggregation methods similar to GraphSAGE, we aggregate the features from its depth $k$ neighborhood, into a single vector, as follows:

$$h_v^{(k)} = \text{AGGREGATE}(\{h_u : \forall u \in s(\mathcal{N}_v^{(k)})\})) \tag{1}$$

where $s(x)$ gives a sample of nodes from the neighborhood set $x$; AGGREGATE is a feature aggregation method similar to GraphSAGE and $k \in \{1, \ldots, K\}$, where $K$ is the max-depth until which we look. This learned embedding $h_v^{(k)}$ captures the node $v$'s neighborhood information from depth $k$.

Now, for each node $v$, the learned embedding at different depths $\{h_v^{(k)}, \forall v \in V\}$ are combined using a Long Short Term Memory (LSTM) [Hochreiter & Schmidhuber (1997)] cell, to give the final node embedding $h_v$ for $v$:

$$h_v = \text{LSTM}(\{h_v^{(K)}, h_v^{(K-1)}, \ldots, h_v^{(1)}\}) \tag{2}$$

This training ensures feature propagation from higher depths of the node's neighborhood to the node. Finally, the learned vectors $\{h_v, \forall v \in \mathcal{V}\}$ are fed to a fully-connected layer for downstream node classification task. Figure 1 gives an overview of the proposed model.

### 2.3 TRAINING

We train our model for supervised multi-class node classification task (transductive as well as inductive) using classification cross-entropy as the loss. The model is trained end-to-end using the same training objective. For multi-label, multi-class classification we use binary cross entropy for each class. To boost training we apply this loss function at each time step of the LSTM instead of only the last layer.

All models were implemented in TensorFlow with the Adam optimizer with learning rate of $0.01$ or or $0.05$. For all the experiments, we consider nodes in the 4-neighbourhood of the center node, i.e., we go upto depth $K = 4$. We learn 128-dimensional node embeddings in each of the experiments. We vary the number of nodes to be sampled for each level of proximity from 5 to 50 depending on the network's density. We use ReLU (rectified linear unit) non-linearity as activation functions in all our variants.

## 3 EXPERIMENTS

In this section, we compare our proposed model against GraphSAGE, on transductive as well as inductive tasks, on four real-world benchmark datasets (Cora, Pubmed and PPI and Citeseer), and show that our model betters or achieves the state-of-the-art performance across all of them.

### 3.1 DATASETS

- **Transductive Setup:** We use Cora, Pubmed and Citeseer citation network datasets for evaluating our model on transductive setup. Nodes in these networks are documents and edges denote the citation relation between them. Node features are the bag-of-words representations of documents. The stats about these networks are shown in Table 1. Here the nodes have access to the features of the nodes in the test and validation set. But the network was not trained on these nodes.

- **Inductive Setup:** The inductive learning requires learning of role of different type of nodes in the network. Here, features of the nodes in the test set and validation set were not used in any form while training. We use protein-protein interaction (PPI) dataset [Hamilton et al. (2017)] consisting of total 24 graphs corresponding to different human tissues. We use 20 graphs for training, 2 for testing and 2 for validation. This is a multi-label classification task.

|       | Cora | Pubmed | PPI | Citeseer |
|-------|------|--------|-----|----------|
| Task  | T    | T      | I      | T    |
| $\vert\mathcal{V}\vert$ | 2708 | 19717 | 56944  | 3312 |
| $\vert\mathcal{E}\vert$ | 5429 | 44338 | 818716 | 4715 |
| $\vert\mathcal{F}\vert$ | 1433 | 500   | 50     | 3703 |
| $\vert\mathcal{L}\vert$ | 7    | 3     | 121    | 6    |

Table 1: Datasets summary. In task T stands for Transductive and I for inductive.

|        | PPI$^{I}$ | Cora  | Pubmed | Citeseer |
|--------|-----------|-------|--------|----------|
| GS-M   | 0.592     | **0.878** | 0.862  | 0.709    |
| GS-G   | 0.5       | 0.854 | 0.812  | 0.688    |
| Ours-M | **0.706** | 0.836 | **0.887** | 0.722  |
| Ours-G | 0.671     | 0.87  | 0.876  | **0.725** |

Table 2: Micro-F1 scores for node classification. GS stands for GraphSAGE. -M and -G are the mean and GCN variants. $^{I}$: Inductive task.

### 3.2 RESULTS

The results of our models with different aggregators are compared with equivalent GraphSAGE models in Table 2. Although the results are only compared to only 2 of the aggregators, our results are better than the best performance from the 4 aggregators.

**Transductive Setup:** On Cora datasets, our proposed - GCN variant attains $\sim 2\%$ improvement over GraphSAGE - GCN. While on Pubmed, we get $\sim 8\%$ improvement with our GCN variant over GraphSAGE - GCN. On Citeseer also, we show 2% improvement in the performance over GraphSAGE.

**Inductive Setup:** For inductive setup on PPI datasets, we obtain an impressive 13.6% improvement over GraphSAGE - mean variant and achieves state-of-the-art 67.4% f1-score.

## 4 CONCLUSION

In this work, we presented an hierarchical feature aggregation model for network embedding which leverages higher-order neighborhood features information more explicitly. We showed good improvements over state-of-the-art approaches.

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
