# OpenReview forum: "Network Embedding using Hierarchical Feature Aggregation"
_ICLR.cc/2018/Workshop — Reject_

### Official Review · AnonReviewer1 · 2018-03-02
**Simple extension of GCNs**

**Rating:** 6
**Confidence:** 4

**Review:**

SUMMARY.

The paper presents a marginal extension of Graph convolutional networks.
The authors propose to aggregate the feature vectors of each GCN step using an LSTM in order to better capture information coming from distant neighbours.
The proposed extension showed to be effective on several datasets.


----------

OVERALL JUDGMENT
The paper is clear and well written.
The proposed extension improve the performance of already state-of-the-art methods.
On the negative side, this approach looks like a glorified residual connection between GCN representation.
It would be interesting to see the behaviour with just a sum of the node representations at each layer, with and without loss function at each step.
It would also be interesting to see the results of the LSTM aggregation without loss function at each step.
These experiments would give a more precise idea of the reason for improvement of the model.


Lastly, the first of the two key limitations is a bit fuzzy to me, could the authors be more clear?

----------

PROS
Simple
Good experimental results

CONS
Marginal extension of GCNs
Not clear what is the impact of some proposed modelling decisions

---

### Official Review · AnonReviewer3 · 2018-03-10
**Uses an LSTM to aggregate information from different hop neighbourhoods**

**Rating:** 5
**Confidence:** 4

**Review:**

1. This paper aggregates embeddings for different hop neighbourhoods for a node by using an LSTM. Though the idea is interesting, there is no intuition why using an LSTM is a good idea. Please better explain this.
2. The authors claim that "the propagation from nodes at higher-depth proximity gets averaged-out multiple times before reaching the centre node". Although, there is no evidence that this indeed is the reason for poor performance when using larger neighbourhoods. Please give either some intuition or empirical proof for this.
There are other unsubstantiated claims below:
3. "To boost training we apply this loss function at each time step of the LSTM instead of only the last layer." Please explain this. What is the performance if you do not do this. In short, please do an ablation study disentangling the improvement in performance by using an LSTM and this trick.
4. "We vary the number of nodes to be sampled for each level of proximity from 5 to 50 depending on the network’s density". This seems to be quite low for a size-k neighbourhood (for k >= 2). Please justify this choice. Is this because of computation reasons? What is the effect in performance if you further increase this?
In summary, although, the basic idea is interesting, it is not very innovative. Further, the lack of ablation studies and justification for some design choices puts this paper below the borderline.

---

### Decision · Program_Chairs · 2018-03-20
**ICLR 2018 Workshop Acceptance Decision**

**Decision:**

Reject

**Comment:**

Based on the reviews, this paper has not been accepted for presentation at the ICLR workshop. However, the conversation and updates can continue to appear here on OpenReview.